# GC–MS Untargeted Analysis of Volatile Compounds in Four Red Grape Varieties (*Vitis vinifera* L. cv) at Different Maturity Stages near Harvest

**DOI:** 10.3390/foods11182804

**Published:** 2022-09-11

**Authors:** Xiaobo Gu, Xue Zhang, Keqing Wang, Xi Lv, Ruyi Li, Wen Ma

**Affiliations:** 1School of Food & Wine, Ningxia University, Yinchuan 750021, China; 2Wine Institution of Ningxia Region, Yinchuan 750021, China

**Keywords:** wine grape, volatile compounds, aroma maturity, harvest

## Abstract

Grape volatile compounds directly determine the aroma quality of wines. Although the aroma profile of grapes evolved greatly at different maturity stages, there were less considerations for aroma status when determining grape harvest time. In the present study, several maturation indicators, namely, sugars/acids ratio, free volatile compounds, bound volatile compounds and IBMP (3-isobutyl-2-methoxypyrazine) content were monitored in four red wine grape varieties (*Vitis vinifera* L. cv Cabernet Sauvignon, Cabernet Gernischet, Cabernet Franc and Merlot) near harvest time (42 days) in Ningxia, China. The results showed that the highest sugars/acids ratio was reached on day 21 and day 28 for Merlot and the other three varieties, respectively. For both free and bound volatile compounds, the content of carbonyl compounds decreased continuously in the process of ripening. The contents of free alcohols, esters and terpenes increased in the ripening stage and decreased in the stage of over-ripening. The accumulation of favorable bound aroma compounds peaked at day 35. The content of IBMP presenting a green smell in all four varieties descended continuously and kept steady from day 28. Therefore, the present findings revealed that the best aroma maturity time of four studied grape varieties was later than the sugars/acids ratio in Ningxia region. Aroma maturity should be taken into account during harvest time determination.

## 1. Introduction

The wine volatile profile consists of numerous aromatic components derived from grapes, wine fermentation and aging. It contributes greatly to the flavor features of wine [1]. Accumulation of volatile compounds in grapes greatly influences wine aroma quality, which is determined by the varietal characteristics of grapes and can also be altered by environmental conditions, vineyard management practices and berry maturity [2,3]. Grapes can contribute aromatic compounds in free or bound forms to wines, which involve monoterpenes, C13-norisoprenoids, C6 compounds and methoxypyrazines (MPs) [4,5,6]. The free aromatic compounds in the berry can be directly transferred into the wine during the winemaking process, while the bound forms, mainly glycosidic precursors, can be converted into volatile compounds during the winemaking process through enzymatic hydrolysis or acid hydrolysis to contribute to the wine aroma [7,8,9].

Grape maturity can be followed using various indicators, e.g., sugars, pH, tartaric acid, sugars/acids ratio, phenolic compounds, seed tannins, berry detachment and aroma. Aroma maturity is a decisive factor in determining the flavor quality of wine. Since the aromatic compounds and their precursors in grapes are synthesizing continuously during berry development, the varietal aromatic components accumulate differently at diversified maturity stages [10,11]. It has been discovered that the maturity of the berry affects the content of monoterpenes, alcohols, aldehydes, etc. The content of monoterpenes in the berry starts to increase after fruit set, decreases near the veraison and then starts to accumulate again and increases as the berry ripens and sugar accumulates; the content of alcohols in the berry also maintains an increasing trend during the ripening process after veraison, while aldehydes show a constant decrease in content [12,13]. During ripening, aroma glycosides also accumulate, and the content of aroma glycosides increase from veraison and reach a peak at ripening time, but decrease during the over-ripening of the grapes [8,14,15]. In addition, IBMP is an aroma compound that is highly correlated with berry maturity because it usually brings green pepper and grass to the wine and has a very low sensory threshold of 10 ng/L [16,17]. IBMP in grapes is synthesized and accumulates before veraison, and its level decreases continuously during the subsequent ripening process [18]. Although many studies have revealed the changes of aroma compounds during grape ripening, the influence of different regions and grape varieties on the accumulation of volatile compounds in grapes cannot be neglected.

Ningxia Helan Mountain’s east foothill is located between the alluvial plain of the Yellow River and the alluvial fan of the Helan Mountains in Ningxia, China, at 37°–39° N and 105°–106° E. With the advantages of abundant sunshine, heat, low precipitation and a large temperature difference between day and night, it has become one of the largest wine-producing regions in China [19]. The majority of wine grape varieties grown in the Ningxia region are *Vitis vinifera* L. cv, which were introduced from Europe. *Vitis vinifera* L. cv Cabernet Sauvignon, Cabernet Gernischet, Cabernet Franc and Merlot are the most planted red wine grape varieties in this region. Cabernet Gernischet (*Vitis vinifera* L.) is a wine grape variety widely cultivated in China, although it is very rare in other producing areas in the world. Although several vineyards and wineries have been initiated in the Ningxia region, no information on the evolution of their volatile profiles during grape ripening has been reported. Improving the aromatic quality of wine by determining the best harvest time and winemaking strategies (maceration and blending) remains challenging.

Therefore, in order to improve the aromatic quality of wine by determining the best harvest time, the present study aimed to characterize the evolution of both free and bound volatile profiles of the four most common red wine grape varieties during ripening and to reveal the regulation of grape aroma maturation in Ningxia Helan Mountain’s east foothill of China.

## 2. Materials and Methods

### 2.1. Chemicals and Reagents

Analytical grade reagents NaCl and NaOH were obtained from Guangnuo Chemical Technology (Shanghai, China); chromatographic grade reagents including ethanol, methanol, dichloromethane; polyvinylpolypyrrolidone (PVPP) and d-gluconic acid lactone with ≥99% purity were purchased from Aladdin Reagent (Shanghai, China); standards of C8–C40 n-alkanes, IBMP, Isopropyl-methoxypyrazine (IPMP), sec-butyl-methoxypyrazine (SBMP) with ≥99% purity and nonanal, acetoin, 1-octanol, isopentyl acetate, linalool with ≥98% purity were from Sigma-Aldrich (St. Louis, MO, USA); standard of 4-methyl-2-pentanol with ≥98% purity from Tokyo Chemical Industry (Tokyo, Japan). Cleanert PEP-SPE columns (200 mg/6 mL) were from Bonna-Agela Technologies (Tianjin, China). Rapidase AR 2000 glucosidase was supplied by Creative Enzymes (New York, NY, USA).

### 2.2. Vineyard and Grape Samples

The Cabernet Sauvignon, Cabernet Gernischet, Cabernet Franc and Merlot samples were picked from the vineyard located in Qingtongxia subregion of Ningxia Helan Mountain’s east foothill (105°54′44″ E, 38°2′30″ N, elevation of 1036 m). The vines were planted in 2010. The overall yield of the vineyard was about 5250 kg/ha.

The grapes of all four varieties were harvested near ripeness, with seven picking dates of 3 September, 10 September, 17 September, 24 September, 1 October, 8 October and 15 October 2020 (hereafter abbreviated as D0, D7, D14, D21, D28, D35 and D42). The temperature and rainfall of the vineyard during the sampling period are presented in Appendix A. For each variety, the grapes were collected by removing the marginal rows of the planting area and randomly selecting three rows in the remaining area as triplication. Then, 0.5 kg samples were picked from each row, for a total of 1.5 kg of each variety. The samples we collected were intact, with similar size and without dehydration or microbial spoilage. The samples were stored at 0 °C and immediately taken back to the laboratory. They were stored in a −80 °C refrigerator before further analysis.

### 2.3. Physicochemical Parameters

The berry weight was measured in 100 berries, and then 50 berries were pressed to obtain grape juice. The juice was centrifuged and used for reducing sugar, titratable acidity and pH analysis. Reducing sugar was determined after 10 times dilution and using titration with Fehling reagent. Titratable acidity was determined using standardized 0.1 N NaOH and expressed as tartaric acid equivalents. pH was measured using a pH meter [20]. All the analyses were conducted in triplicate.

### 2.4. Free Aroma Volatile Components Analysis

Maceration of free volatile components: 100 grape berries were randomly selected, completely frozen in liquid nitrogen and then ground into powder in a high-speed crusher (Zhejiang Rhodiola Co., Ltd., Zhejiang, China). Subsequently, 15 g of powder was carefully weighed and placed in a 50 mL centrifuge tube and blended with 0.15 g PVPP and 0.1 g d-gluconic acid lactone under liquid nitrogen protection. The flesh was macerated at 4 °C for 4 h before being centrifuged at 7104 g at 4 °C for 10 min, and the clear juice sample was obtained [21]. Each maceration process was performed in duplicate.

Headspace–solid-phase microextraction (HS–SPME) conditions refer to Lan et al., with slight modification [21]: adsorption of free volatile compounds was directly conducted with HS–SPME using a CTC PAL RSI 85 (CTC Analytics, Zwingen, Switzerland) autosampler equipped with 1 cm DVB/CAR/PDMS 50/30 μm SPME fibers (Supelco, Bellefonte, PA, USA). The SPME fibers were first activated at 250 °C for 10 min. Then, a 20 mL headspace vial containing 5 mL of juice sample, 2.0 g NaCl and 10 μL 4-methyl-2-pentanol (internal standard) was moved to a heating equipment shaker and stabilized at 40 °C and 400 rpm for 5 min. Subsequently, the SPME fibers were inserted into the headspace vial for adsorption at 40 °C and 400 rpm for 30 min with a piercing depth of 3.4 cm. Finally, the SPME fibers were inserted into the injection port and desorbed at 240 °C for 10 min.

Gas chromatograph–mass spectrometer (GC–MS) analysis was performed on an Agilent 7890B GC in tandem with an Agilent 7000D MS. The separation was performed on a DB-WAX capillary column (30 m × 0.25 mm × 0.25 μm, Agilent, J & W Scientific, Folsom, CA, USA). The GC temperature program was taken from [22] and was as follows: an initial temperature of 40 °C was maintained for 5 min, and increased by 3 °C/min to 97 °C/min, held for 5 min, followed by heating to 120 °C at a rate of 2 °C/min, next at 3 °C/min speed to 150 °C, and finally increased by 8 °C/min to 220 °C and held for 10 min. The whole process was carried out by a helium carrier with a flow rate of 1 mL/min. The MSD transfer line heater was set to 250 °C. The temperatures of the ion source and quadrupole were 230 °C and 150 °C, respectively. The mass detector was operated in full scan mode (*m*/*z* 30–350) with electron ionization (EI) mode at 70 eV. Each analysis was performed in duplicate.

### 2.5. Bound Volatile Components Analysis

Extraction of bound volatile components: for the extraction steps of glycosidically bound aroma compounds refer to Chen et al. [23]. Isolation of glycosidically bound aromatic precursors was conducted using Cleanert PEP-SPE resins (Bonna-Agela Technologies Co., Ltd., Tianjin, China), which was previously conditioned with 10 mL of methanol and 10 mL of ultrapure water (UPHQ-I-90T water system, Ulupure Technology, Chengdu, China). An amount of 5 mL of centrifuged juice supernatant was gravitationally passed through a solid-phase extraction (SPE) column. The column was rinsed with 5 mL of ultrapure water to remove sugars, acids and most other polar compounds and then washed with 5 mL of dichloromethane to eliminate free fractions. The precursors were eluted with 10 mL of methanol. The methanol extract was evaporated to dryness under a nitrogen stream and redissolved in 10 mL of citrate–phosphate buffer solution (0.2 M, pH 2.5). Enzymatic hydrolysis was conducted in an incubator with the addition of 100 mL of AR2000 (Rapidase, 100 g/L) at 40 °C for 24 h. Each extraction process was performed in duplicate.

SPME and GC–MS conditions were the same as in Section 2.4.

### 2.6. IBMP Analysis

The extraction method of IBMP in berry was as in Lei et al. [24]. The frozen whole berries (15 g) were weighed and deseeded, ground into a fine powder in liquid nitrogen and placed in a 50 mL plastic centrifuge tube. An aliquot of 2 mM NaF solution (5.0 mL) was added to the powder in the tube. The solution was homogenized with a vortex mixer until smooth. After homogenization, the temperature of the solution was maintained below 2 °C. The suspension was centrifuged at 8000 rpm at 4 °C for 10 min. An aliquot (5.0 mL) of the supernatant was transferred to a 20 mL screwcap headspace vial containing 2.0 g of NaCl. The extraction method of IBMP from the grape stem, pulp, skin and seed is similar to that of whole berries, except that each extraction used only 10 g of powder from different parts and added 10 mL of 2 mM NaF solution.

HS–SPME conditions were the same as in Section 2.4. GC–MS analysis for IBMP used the method of Botezatu et al. [25] with slight modifications. GC–MS analysis was performed on an Agilent 7890B GC in tandem with an Agilent 7000D MS. The separation was performed on a DB-WAX capillary column (30 m × 0.25 mm × 0.25 μm, Agilent, J & W Scientific, Folsom, CA, USA). The GC temperature program was as follows: an initial temperature of 40 °C was maintained for 10 min, and increased by 10 °C/min to 100 °C/min, followed to 140 °C at a rate of 3 °C/min, and finally increased by 25 °C/min to 230 °C and held for 3 min. The whole process was carried out by a helium carrier with a flow rate of 1 mL/min. The MSD transfer line heater was set to 250 °C. The temperatures of the ion source and quadrupole were 230 °C and 150 °C, respectively. The mass detector was operated in selected ion monitoring (SIM) mode with EI mode at 70 eV. SIM was used at mass channels of *m*/*z* 94 and 124 for IBMP. Peak areas of the ion *m*/*z* 124 were used for quantification.

IBMP standard curve establishment: IBMP standard was first diluted to 1 mg/L with ethanol, and then a juice sample (IBMP supernatant extracted from berry) was used to make a second dilution of 1 μg/L stock solution. IBMP juice sample solutions (0, 2, 5, 10, 15, 30, 50 ng/L), 5 mL of each concentration was added to a 20 mL headspace bottle with 2.0 g NaCl and 10 μL of internal standard (4-methyl-2-pentanol), and two parallel sets were made for each concentration. The peak area of the IBMP quantitative ion (*m*/*z* 124) was standardized using the internal standard characteristic ion peak area (*m*/*z* 45) and the content was calculated to produce the standard curve. The validation of the method was verified by configuring IBMP juice sample solutions at concentrations of 20 ng/L and 35 ng/L, with two replicates for each concentration. The specific parameters of the standard curves are shown in Appendix A.

### 2.7. Qualitative Analysis and Quantification of Volatile Compounds

The mass spectra were searched using the NIST 17 standard mass spectrometry library, and the retention index (RI) was calculated based on the retention times of C8–C40 n-alkanes under the same conditions. The RIs were compared with those of the literature using the same column, and the same compound was identified as having an RI difference of five or less. The identification of IBMP was by pure standard [26].

The concentration of the compound was calculated by multiplying the ratio of the characteristic ion peak area of the compound and the characteristic ion peak area of the internal standard by the concentration of the internal standard; the concentration of IBMP was calculated using the standard curve shown in Section 2.6.

### 2.8. Statistical Analysis

All statistical analyses were performed by R 3.6.3 software [27]. One-way analysis of variance (ANOVA) was carried out to evaluate significant differences (*p* < 0.05) among physicochemical parameters of berries at different maturity stages, and least significant difference (LSD) post hoc test was used. Cluster analysis was performed to explore the evolution of free and bound volatile compounds by ‘pheatmap’ package of R environment. Principal components analysis (PCA) was conducted to present the distribution of volatile compounds in the four wine grape varieties, using ‘FactoMineR’ package with confidence interval of 95%.

## 3. Results and Discussion

### 3.1. Physicochemical Parameters

Berry development stages were observed by measuring total sugars, titratable acidity, pH, weight per 100 berries and sugars/acids ratio. As shown in Table 1, the sugars/acids ratio of Cabernet Sauvignon, Cabernet Gernischet and Cabernet Franc increased continuously from D0 to D28, reached the maximum value at D28, and then decreased until D42. Meanwhile, the peak of the Merlot sugars/acids ratio appeared at D21. Hence, according to sugars/acids ratios, the ripen date of Cabernet Sauvignon, Cabernet Gernischet, Cabernet Franc and Merlot were D28, D28, D28 and D21, respectively.

During the ripening process, the total sugars of all four grape varieties experienced a significant rise. The total sugars of Cabernet Sauvignon rose from 173.8 g/L to 240.8 g/L during the first 28 days, with an increase of 67 g/L. At the same time, Cabernet Gernischet and Cabernet Franc showed an increase of 48 g/L and 40.9 g/L, respectively. The total sugars of Merlot experienced an increase of 46.8 g/L from D0 to D21 (its sugars/acids maturation time). During the over-ripening period, Cabernet Sauvignon kept accumulating sugars, while the other three varieties experienced a slight decrease. Similar variation trends were also observed in weight per 100 berries in the four varieties. During berry ripening, they rose, while in the over-ripening period, they decreased. This may be due to the fact that temperature is high and precipitation sufficient, which contribute to fruit development and sugar accumulation during the ripening process; then, the temperature and precipitation decreased, which cause both carbohydrate depletion and shrinking of berries during the over-ripening period (Appendix A and Table 1).

During the ripening process, the titratable acid of the four grape varieties kept decreasing, with a decrease of 1.4–3.5 g/L. The same trend was also observed for Merlot in D28–D42 due to the inhibition of acid synthesis and the increase in catabolism during berry ripening and the dilution effect caused by the increase in berry size. During the over-ripening process from D28 to D42, a small increase in the titratable acid of Cabernet Sauvignon, Cabernet Gernischet and Cabernet Franc was observed, which may be caused by water loss [28].

In sum, before the ripening date, total sugars, pH, berry weight and sugars/acids ratio increased, while total acid concentration decreased with time. During over-ripening, pH, total sugars and titratable acid showed different fluctuations, berry weight decreased consistently, accompanied by different levels of organic matter depletion and water loss.

### 3.2. Evolution of the Concentration of Volatile Compounds during Grape Ripening

A total of 36 free volatile compounds and 40 bound volatile compounds, including alcohols, carbonyls, esters, terpenes and acids were detected in the grapes using HS–SPME–GC–MS analysis (data shown in Appendix A).

#### 3.2.1. Free Volatile Compounds

The accumulation pattern of free volatile compounds during the ripening process in the four varieties of wine grapes is shown in Figure 1, which are presented by hierarchical cluster analysis. The free volatiles can be divided into two categories according to different trends, with the first category showing a decrease in concentration during ripening, the second category showing an increase in concentration during ripening.

The aromas of wine grapes in the D0–D7 period were dominated by carbonyl compounds and showed a decreasing trend during grape ripening. Among the first category compounds, C6 aldehydes were predominant, including 1-hexenal, 3-hexenal, 2-hexenal, (E)-2-hexenal and (E,E)-2,4-hexadienal. These compounds all had the maximum content in the D0–D7 period, which is just the source of green aromas in the unripe berry. During grape maturation, C6 aldehydes are formed from fatty acids. Fatty acids such as linoleic acid and linolenic acid are oxidized by lipoxygenase (LOX) and lipid hydroperoxide lyase to produce C6 aldehydes. Furthermore, C6 aldehydes are reduced to C6 alcohols by alcohol dehydrogenase (ADH), resulting in a reduction of C6 aldehydes during grape berry ripening [29]. It was suggested that the ratio of C6 aldehydes to C6 alcohols could be used to determine the ripeness of grapes from an aromatic point of view and that this ratio continued to decrease during the ripening process [30,31]. The sensory threshold of C6 alcohols is higher than that of C6 aldehydes and has less effect on the aroma. This conversion reduces the green flavor during grape berry ripening. In contrast, C6 alcohols were mostly classified as the second category of compounds, including n-hexanol, (E)-3-hexenol, (Z)-3-hexenol and (Z)-2-hexenol, all of which tended to increase during ripening, with some decreasing during the over-ripening process. Differences in C6 alcohols in different varieties of wine grapes occurred mainly during over-ripening, for example, n-hexanol and (Z)-3-hexenol peaked at D42 in Cabernet Sauvignon and Cabernet Gernischet, while n-hexanol in Cabernet Franc reached a maximum at D21, after which the levels decreased. It was found that C6 alcohols increased and then decreased rapidly during water loss from grapes at over-ripening, the content of C6 compounds was proportional to the activities of ADH and LOX [32], while LOX and ADH activity levels varied with the rate and amount of water loss [33]. In addition, the degree of water loss in the early stages of over-ripening in different varieties of grapevines is susceptible to soil conditions, climate and genetic factors, leading to different levels of water loss in different varieties at the same time and thus having different levels of LOX and ADH activity, while varieties with less water loss continue to synthesize C6 alcohols at the early stages of over-ripening, leading to differences in the levels of C6 alcohols at the early stages of water loss in different varieties.

As can be seen from Figure 1, wine grape aroma was dominated by alcohols, esters and terpenes in the D28–D42 period. Most of the alcohols, esters and terpenes were clustered together and showed an overall upward trend during berry ripening, which belonged to the second category. Among them, the contents of benzyl alcohol, phenylethyl alcohol, theaspirane and butyl acetate showed an upward trend during the ripening of berries from all four cultivars and were able to enhance the floral and fruity notes in ripened berries. However, not all alcohols, esters and terpenes changed equally during ripening in different cultivars. For example, 2-heptanol and methyl heptenone increased gradually during ripening in Cabernet Sauvignon, Cabernet Franc and Merlot berries, whereas they decreased in Cabernet Gernischet. Ethyl caprylate, which increased during ripening in Cabernet Gernischet, Cabernet Franc and Merlot, decreased in Cabernet Sauvignon. This variability in the accumulation of aroma compounds of different varieties may be responsible for the distinction among grape varieties.

#### 3.2.2. Bound Volatile Compounds

Through the cluster analysis of Figure 2, it was found that, compared with free volatile compounds, most of the bound volatile compounds showed an increase during grape ripening. The bound volatiles can also be divided into two categories according to different trends, with the first category showing a decrease in concentration during ripening and the second category showing an increase in concentration during ripening.

In the first category, the bound carbonyl compounds (octanal, nonanal and decanal) showed a continuous decrease during ripening in different varieties, the same as their free fraction. The two bound C6 aldehydes detected showed quite opposite trends, (E)-2-hexenal showing a continuous decrease in the four varieties, while hexanal increased. This may result from the enzymatic hydrolysis or hydrolysis of the glycoside bound precursors in the ripe berry during the winemaking stage [17,21], but this result is negligible compared with the significant decrease in the free state of hexanal; the other carbonyl species, methyl isobutyl ketone, benzaldehyde and 2,4-dimethylbenzaldehyde showed the same trend as that of hexanal, and all were second category compounds.

Among the bound alcohols, besides (Z)-2-hexenol and (Z)-3-hexenol, some alcohols showed the same trend as the alcohols in the free fraction in the ripening process, and all of them showed an increasing trend and were second category compounds, whereas (Z)-3-hexenol showed a continuous decrease in the ripening process of different varieties of berry and was a first category compound. (Z)-2-Hexenol showed different trends among different varieties; its trends in Cabernet Sauvignon, Cabernet Franc and Merlot were the same as most of the alcohols, but it only showed a decreasing trend in the ripening process of Cabernet Gernischet.

The accumulation of bound terpenoids followed the same trend as the accumulation of their free states, belonged to second category compounds, with the majority of bound terpenoids increasing in content during berry ripening (cedrol, (Z)-ocimene, (E)-ocimene, terpinolene, 1,3-octadiene). As can be seen from the graph, the peaks of terpenoids mostly occurred during the D28–D35 period, with some decrease in content during the subsequent one to two weeks of over-ripening. A similar result was reported in “Moscato biano” and “Aleatico” varieties. During the over-ripening process, a decrease in the content of monoterpene glycoside precursors was also observed [34]. Lan also showed the same phenomenon in “Beibinghong” grapes during over-ripening [21].

### 3.3. Distribution of Volatile Compounds in the Four Wine Grape Varieties

#### 3.3.1. Free Volatile Compounds of the Four Wine Grape Varieties

Free volatile compounds of the four wine grape varieties were analyzed using PCA to investigate the effect of varieties on aroma profile. As shown in Figure 3a1, the first (44.22%) and second (23.14%) components explained 67.36% of the total variance. The over-ripen samples of different varieties (D35 and D42) were well separated from the early samples by PC1. The main contributors to PC1 were aldehydes (hexanal, (E)-2-octenal, 2,4-heptadienal, 2.4-hexadienal, etc.) and acetophenone (Figure 3a2). These compounds and the over-ripen samples have opposite trend vectors in PC1, indicating that the contents of these compounds were negatively correlated with the ripeness of grapes.

In Figure 3a1, different varieties were also well separated by PC1 and PC2. The samples of Cabernet Sauvignon were distributed in the first and second quadrants. According to the loading plot (Figure 3a2), more types of free volatile compounds contributed to the aroma of Cabernet Sauvignon than the other three varieties. For Cabernet Sauvignon, samples picked at D0–D7 were mainly located in the first quadrant, contributed by hexanal, acetophenone, (E)-2-Octenol, etc. D14–D28 distributed at the intersection of the first quadrant and the second quadrant were contributed by benzyl alcohol, 1-Hexenol, (Z)-3-Hexenyl acetate and other compounds. D35–D42 were contributed by (E)-3-hexenol, acetophenone, benzyl alcohol and other compounds and distributed in the second quadrant. Cabernet Gernischet samples collected at various dates were mainly located in the fourth quadrant, with contributors of octanal, 3-hexenal and decanal, etc. The sample distribution of Cabernet Franc was close to that of Cabernet Gernischet at the center of the coordinate axis. The Merlot samples were distributed in the third quadrant in Figure 3a1, far apart from the three other varieties. These samples were characterized by the compounds of butyl acetate, ethyl caproate, ethyl caprylate, methyl heptanone and methyl isobutyl ketone.

#### 3.3.2. Bound Volatile Compounds of the Four Wine Grape Varieties

Bound volatile compounds of the four wine grape varieties were also analyzed using PCA to investigate the effect of varieties on aroma profile. As shown in Figure 3b1, the first (31.11%) and second (29.98%) components explained 61.09% of the total variance. The over-ripen samples of different varieties (D35 and D42) separated well from the samples in earlier stages by PC1. Bound volatile compounds of benzyl alcohol, α-Terpineol and other compounds contributed to the positive aspect of PC1, while octanal, methyl myristate and other compounds contributed to the negative aspect of PC1.

Different varieties picked at various dates were well separated by PC1 and PC2 of bound volatile compounds. The samples of Cabernet Sauvignon were distributed in the third and fourth quadrants. Compared with other varieties, the number of volatile compounds that contribute significantly to the aroma of Cabernet Sauvignon berry was larger. Among them, D0 was far away from other samples and was located in the third quadrant. According to Figure 3b2, it can be seen that the contribution of (E)-2-hexenal, terpinolene and other compounds in the third quadrant to the berry aroma of Cabernet Sauvignon during the D0 period was larger than that of other samples. More significantly, Cabernet Sauvignon D14–D42 was mainly located at the intersection of the third and fourth quadrants, which contributed to positively by geranyl acetone, ethyl laurate, α-cedarene, ethyl myristate and methyl heptenone. Cabernet Franc samples were mainly distributed in the negative semiaxis direction of PC1, which was characterized by the bound volatile compounds of 2,4-di-tert-butylphenol, octanal, methyl myristate, (E)-2-hexenal, etc. Different from free volatile compounds, Cabernet Gernischet samples were well separated from Cabernet Franc by bound volatile compounds. Cabernet Gernischet was more related to methyl hexadecanoate, methyl myristate, 2-ethylhexanol and 2,4-dimethylbenzaldehyde. The distribution of Merlot samples in Figure 3b1 is located in the first quadrant, on the positive semiaxis of both PC1 and PC2, with contributors of major terpenes ((E)-ocimene, d-limonene, pseudolimonene, etc.).

### 3.4. Variations in the IBMP Contents of Different Grape Varieties during Maturity

MPs, as an important characteristic aroma in wine, are also important volatile indicators for grape aroma maturation. Because these compounds are present in grapes in low concentration (ppt) and low sensory threshold, it is a challenge to quantify them [35]. In the present investigation, only IBMP was detected in grape samples with quantification above the limit of quantification value. As shown in Figure 4, the content of IBMP in the four varieties of wine grapes decreased consistently as the grape ripened, which was contrary to the change of sugar content. The IBMP content of the four varieties bottomed at the last sampling date with the content below 11 ng/kg; there was a significant difference with the peak concentration, which was consistent with the previous results [36]. Varieties also affected the content of IBMP. In general, Cabernet Gernischet and Cabernet Franc had significantly higher IBMP content than Cabernet Sauvignon and Merlot in the Ningxia region during the whole ripening and over-ripening periods. The peak concentration (18.7 ng/L) was reached by Cabernet Gernischet at D7, which was 16.8 ng/L higher than Merlot (1.9 ng/L) at the same sampling date. These differences may result from both variety diversities and adaptation of various varieties in the Ningxia region.

## 4. Conclusions

This study investigated the evolution of sugars/acids ratio, free and bound volatile compounds and IBMP content near ripening time of four grape varieties (Cabernet Sauvignon, Cabernet Gernischet, Cabernet Franc and Merlot) in Ningxia Helan Mountain’s east foothill. According to data of the sugars/acids ratio, Cabernet Sauvignon, Cabernet Gernischet and Cabernet Franc ripened at almost the same time, which was a week later than Merlot; Merlot more easily accumulated a number of sugars.

For the free volatile compounds, the content of carbonyl compounds decreased continuously in the process of ripening. The contents of alcohols, esters and terpenes increased in the ripening stage and decreased in the stage of over-ripening. Other compounds fluctuated in the process. For the bound volatile compounds, carbonyl compounds and alcohols showed the highest proportion; the contents of terpenes and esters were significantly higher than those of free aroma compounds. Carbonyl compounds decreased in the process of ripening, while alcohols, esters and terpenes increased. Furthermore, esters and terpenes tended to decrease in the process of over-ripening. Other compounds fluctuated in the whole period. Through PCA analysis, it was found that bound volatile compounds were able to distinguish the different grape varieties studied in this paper.

During both the ripening and over-ripening periods, the IBMP of whole berries showed a consistent downtrend in content. IBMP in Cabernet Gernischet and Cabernet Franc was always higher than in Cabernet Sauvignon and Merlot.

To our knowledge, this is the first report to offer an insight on the evolution of the volatile compounds of these four grapes near harvest time in the Ningxia region. The present findings demonstrated that the best aroma maturity time of four studied grape varieties was later than the sugars/acids ratio in Ningxia region. Aroma maturity should be taken into account during harvest time determination. Modification of viticulture practices and wine style designs are encouraged. This may improve the aromatic quality of Ningxia wine by serving as a reference for determining the best harvest time and winemaking strategies.

## Figures and Tables

**Figure 1 foods-11-02804-f001:**
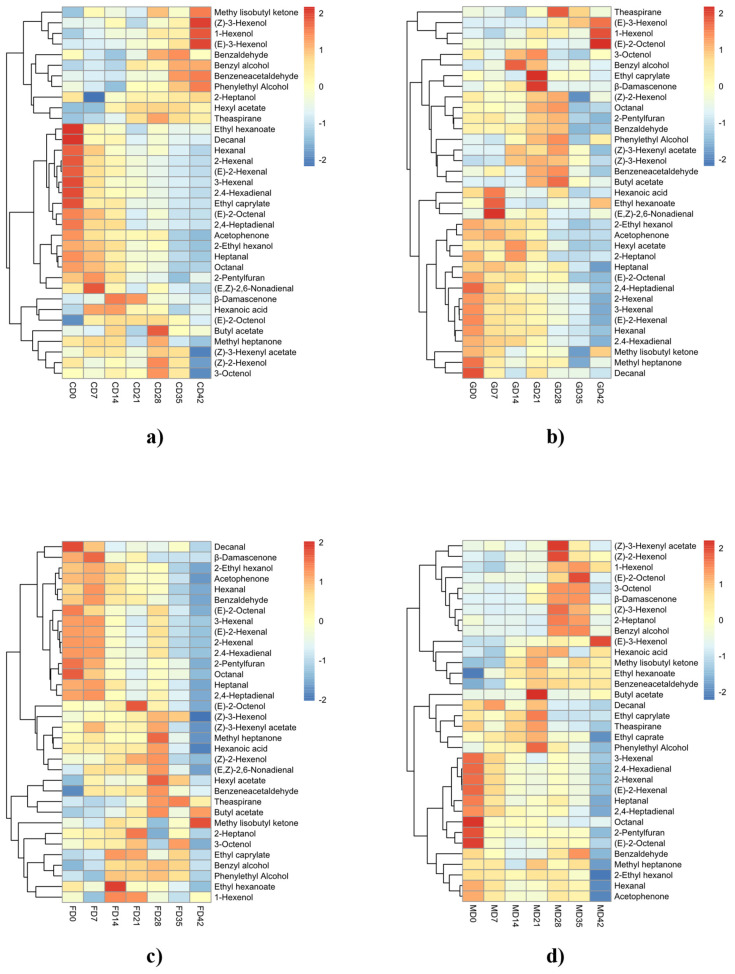
Clustering heat map of free volatile compounds during grape ripening. ((**a**), Cabernet Sauvignon; (**b**), Cabernet Gernischet; (**c**), Cabernet Franc; (**d**), Merlot).

**Figure 2 foods-11-02804-f002:**
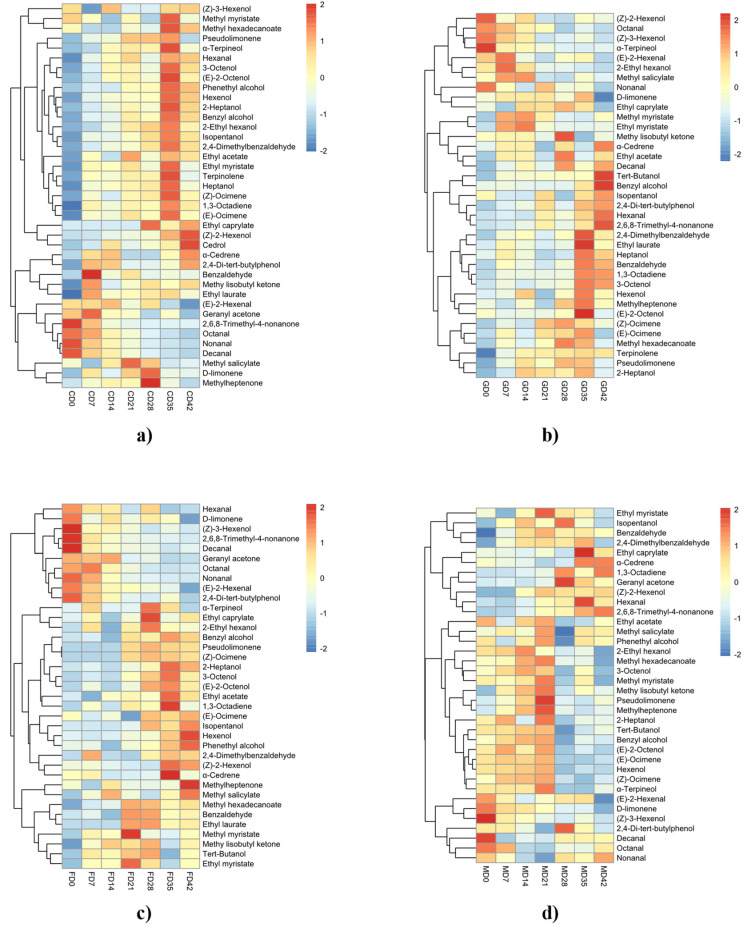
Clustering heat map of bound volatile compounds during grape ripening. ((**a**), Cabernet Sauvignon; (**b**), Cabernet Gernischet; (**c**), Cabernet Franc; (**d**), Merlot).

**Figure 3 foods-11-02804-f003:**
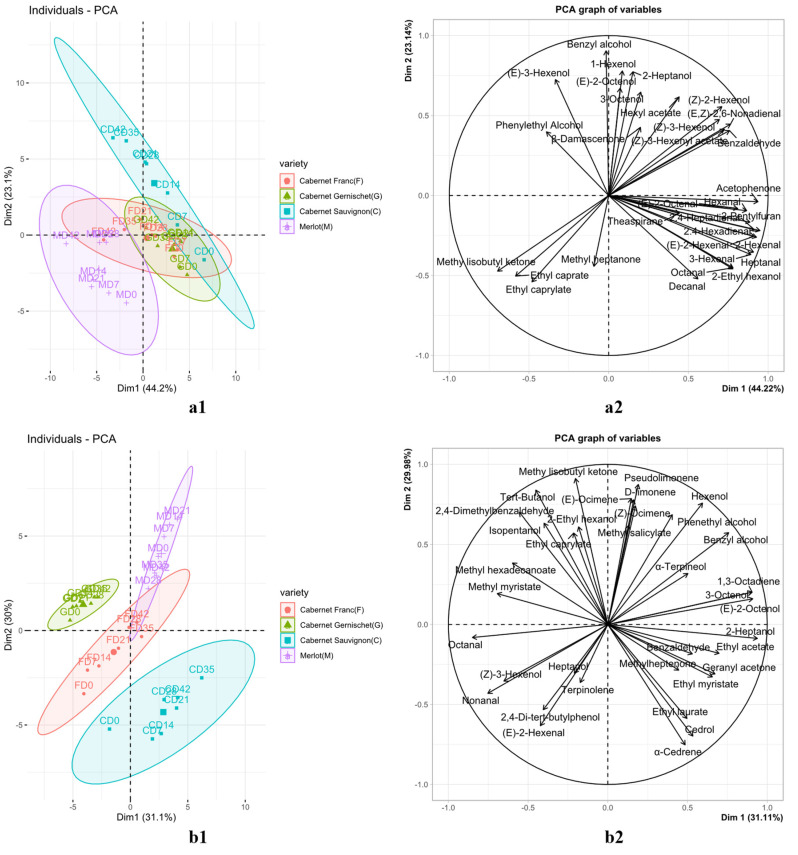
PCA scores plot for samples based on free (**a1**) and glycosidically bound (**b1**) volatile compounds of different varieties; PCA loadings plot of free (**a2**) and glycosidically bound (**b2**) volatile compounds of different varieties.

**Figure 4 foods-11-02804-f004:**
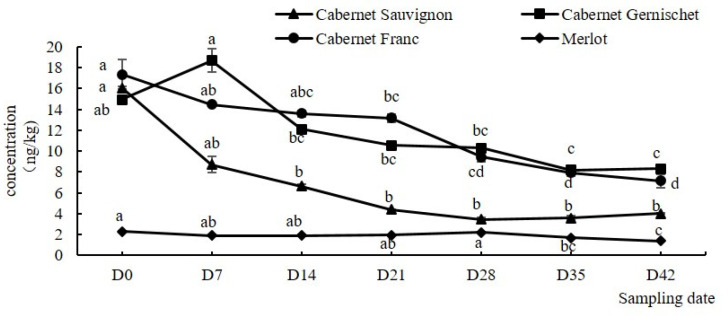
Changes of IBMP during grape ripening. (The ordinate represents the content of IBMP, and the abscissa represents the sampling date explained above).

**Table 1 foods-11-02804-t001:** B Physicochemical parameters of the grapes at different maturity stages.

Physiochemical Parameters	Variety	D0	D7	D14	D21	D28	D35	D42
Sugars/acids ratio	Cabernet Sauvignon	24.69 ± 0.38 ^d^	42.98 ± 0.78 ^cd^	47.75 ± 0.89 ^bc^	54.45 ± 1.05 ^abc^	67.27 ± 0.22 ^a^	62.95 ± 1.37 ^ab^	64.91 ± 1.44 ^ab^
	Cabernet Gernischet	33.28 ± 0.53 ^c^	41.93 ± 1.30 ^bc^	58.27 ± 2.53 ^ab^	68.60 ± 1.17 ^a^	69.49 ± 1.26 ^a^	62.22 ± 1.41 ^a^	65.16 ± 1.55 ^a^
	Cabernet Franc	36.04 ± 0.79 ^c^	45.24 ± 0.00 ^bc^	56.91 ± 1.32 ^ab^	68.73 ± 2.77 ^a^	67.12 ± 1.70 ^a^	53.09 ± 0.86 ^ab^	61.11 ± 1.47 ^abc^
	Merlot	51.00 ± 0.41 ^c^	61.33 ± 0.64 ^bc^	79.56 ± 1.99 ^ab^	95.89 ± 3.16 ^a^	66.65 ± 1.33 ^bc^	68.79 ± 1.77 ^bc^	71.10 ± 2.86 ^bc^
Total Sugar(g/L)	Cabernet Sauvignon	173.82 ± 0.53 ^d^	202.56 ± 3.63 ^c^	210.46 ± 3.92 ^c^	217.99 ± 4.22 ^bc^	240.78 ± 5.01 ^ab^	248.20 ± 5.48 ^a^	255.95 ± 0.00 ^a^
	Cabernet Gernischet	167.05 ± 2.41 ^b^	197.62 ± 3.37 ^a^	210.46 ± 3.92 ^a^	212.24 ± 3.61 ^a^	215.00 ± 3.90 ^a^	209.49 ± 3.90 ^a^	209.49 ± 3.90 ^a^
	Cabernet Franc	183.09 ± 2.53 ^b^	207.69 ± 0.00 ^a^	216.00 ± 0.00 ^a^	220.97 ± 4.22 ^a^	223.96 ± 0.00 ^a^	212.24 ± 3.90 ^a^	215.00 ± 0.00 ^a^
	Merlot	209.16 ± 5.10 ^b^	236.52 ± 2.46 ^a^	253.25 ± 5.51 ^a^	255.95 ± 0.00 ^a^	230.45 ± 4.59 ^ab^	233.70 ± 0.00 ^ab^	237.24 ± 5.01 ^a^
Titratable Acid(g/L)	Cabernet Sauvignon	7.04 ± 0.09 ^a^	4.71 ± 0.09 ^b^	4.41 ± 0.00 ^b^	4.00 ± 0.00 ^b^	3.58 ± 0.09 ^b^	3.94 ± 0.09 ^b^	3.94 ± 0.09 ^b^
	Cabernet Gernischet	5.02 ± 0.09 ^a^	4.71 ± 0.09 ^ab^	3.61 ± 0.09 ^bc^	3.09 ± 0.00 ^c^	3.09 ± 0.00 ^c^	3.37 ± 0.04 ^c^	3.22 ± 0.09 ^c^
	Cabernet Franc	5.08 ± 0.09 ^a^	4.59 ± 0.00 ^ab^	3.80 ± 0.09 ^bc^	3.22 ± 0.09 ^c^	3.34 ± 0.09 ^c^	4.00 ± 0.01 ^bc^	3.52 ± 0.09 ^bc^
	Merlot	4.10 ± 0.00 ^ab^	3.86 ± 0.00 ^ab^	3.18 ± 0.09 ^bc^	2.67 ± 0.09 ^c^	3.46 ± 0.00 ^ab^	3.40 ± 0.09 ^abc^	3.34 ± 0.09 ^bc^
pH	Cabernet Sauvignon	3.48 ± 0.00 ^c^	3.65 ± 0.00 ^bc^	3.66 ± 0.00 ^bc^	3.81 ± 0.01 ^ab^	3.98 ± 0.00 ^a^	3.98 ± 0.00 ^a^	3.98 ± 0.00 ^a^
	Cabernet Gernischet	3.69 ± 0.00 ^b^	3.86 ± 0.00 ^ab^	3.87 ± 0.02 ^ab^	4.04 ± 0.00 ^a^	4.01 ± 0.00 ^a^	4.02 ± 0.00 ^a^	3.92 ± 0.00 ^a^
	Cabernet Franc	3.61 ± 0.01 ^b^	3.80 ± 0.00 ^bc^	3.92 ± 0.00 ^ab^	4.06 ± 0.00 ^a^	4.01 ± 0.00 ^ab^	3.96 ± 0.01 ^ab^	4.00 ± 0.00 ^ab^
	Merlot	3.84 ± 0.00 ^c^	3.94 ± 0.01 ^bc^	3.98 ± 0.00 ^bc^	4.14 ± 0.01 ^a^	4.04 ± 0.00 ^ab^	3.98 ± 0.00 ^bc^	3.98 ± 0.00 ^bc^
Weight per 100 berries(g)	Cabernet Sauvignon	95.08 ± 1.39 ^b^	96.52 ± 1.25 ^b^	99.50 ± 1.18 ^b^	102.38 ± 1.15 ^ab^	112.30 ± 3.35 ^a^	103.12 ± 1.54 ^ab^	97.12 ± 1.47 ^b^
	Cabernet Gernischet	124.20 ± 3.33 ^c^	124.15 ± 4.19 ^c^	127.02 ± 2.05 ^bc^	132.72 ± 2.98 ^abc^	142.92 ± 1.35 ^a^	137.88 ± 2.83 ^a^	137.131 ± 1.73 ^ab^
	Cabernet Franc	119.31 ± 4.46 ^a^	122.64 ± 1.06 ^a^	123.19 ± 2.32 ^a^	125.30 ± 3.02 ^a^	126.25 ± 2.40 ^a^	120.39 ± 6.28 ^a^	106.22 ± 2.35 ^b^
	Merlot	99.82 ± 1.17 ^bc^	101.68 ± 1.03 ^abc^	104.24 ± 1.81 ^ab^	107.94 ± 2.10 ^a^	105.85 ± 1.88 ^ab^	102.68 ± 1.29 ^abc^	95.73 ± 4.63 ^c^

Note: Different letters indicate significant differences between the means of different maturity stages (*p* < 0.05).

## Data Availability

Data are contained within the article or Appendix A.

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
