# Peer review of "GC–MS Untargeted Analysis of Volatile Compounds in Four Red Grape Varieties (Vitis vinifera L. cv) at Different Maturity Stages near Harvest"

_foods, 2022, doi:10.3390/foods11182804_

Round 1

Reviewer 1 Report

The authors present a study that is investigating the volatile and non-volatile aroma profile of selected grape varieties over the ripening period. The study is well designed and the scientific value is good, however, there are a few changes that would improve the overall quality of manuscript before it should be considered for publication.

General comments:

The grape cultivar in the manuscript that is described as "Cabernet Gernischet" is usually spelled "Gernischt" and is actually Carmenere. I am not sure if there are regional differences but if that manuscript could reach a broader audience, I would recommend using the widely known Carmenere name for it.

Besides the overall English editing that is required throughout the manuscript, I would recommend to check the use of present and past tense. General statements and data interpretation that is still true should be present tense. Descriptions of procedures should be past tense.

Specific comments:

Line 19    The abbreviations D21 and D28 are explained only in the methods and materials part. If you use them in the abstract, please explain. Otherwise, use day 21 and day 28 instead here.

Line 46    This is a little unclear. There is very little free aroma in grape berries, most of it is bound in a precursor form. There is really no varietal aroma in grapes for that reason since we cannot perceive bound aroma compounds. This should be clarified here.

Line 102  This goes for the entire manuscript: please avoid abbreviations such as Fig. 1 and use Figure 1 instead.

Line 105  If you randomly selected grapes for sampling, did you assess variability within the vineyard block? If not, the sample size of 1.5 Kg is too small to account for natural site variability.

Line 121  This is a maceration, not an extraction in the traditional sense. 

Line 167  How did you get to the ng/L range without pre-concentration of the analyte? 

Line 202  Please describe the functions that you used in Excel 2019.

Line 225  How does that carbohydrate depletion work in the berry? Could it also be a mix of dehydration and microbial spoilage?

Line 235  Could this also be caused by microbial activity? Did you check for grape health at the different harvest points?

Line 273  If you are using past tense here, it sounds like you did the actual work but as far as I can see, you are referencing a paper. This needs to be a little clearer.

Line 281  Figure 1 and Figure 2 are impossible to read. Please make the heat maps bigger and the font much bigger as well.

Line 296  Not only the distinction of grape cultivars but also growing conditions, regions, etc.

Line 366  Figure 3 is also hard to read. Please make the font bigger.

Line 390  When using abbreviations for the first time, please explain.

Line 430  This was actually already known. The aromatic qualities (and phenolic ripeness) always develops with a delay. Please elaborate on how you expect wineries to use that knowledge (or technology) in order to improve the aromatic qualities of their products.

Author Response

Comments and Suggestions for Authors

The authors present a study that is investigating the volatile and non-volatile aroma profile of selected grape varieties over the ripening period. The study is well designed and the scientific value is good, however, there are a few changes that would improve the overall quality of manuscript before it should be considered for publication.

General comments:

1.The grape cultivar in the manuscript that is described as "Cabernet Gernischet" is usually spelled "Gernischt" and is actually Carmenere. I am not sure if there are regional differences but if that manuscript could reach a broader audience, I would recommend using the widely known Carmenere name for it.

Thanks for your suggestions. Indeed, many studies have shown that they have many similarities in quality, but there is not enough molecular evidence to prove that they are a variety.

2.Besides the overall English editing that is required throughout the manuscript, I would recommend to check the use of present and past tense. General statements and data interpretation that is still true should be present tense. Descriptions of procedures should be past tense.

Thanks for kind remind. We have checked.

Specific comments:

1.Line 19    The abbreviations D21 and D28 are explained only in the methods and materials part. If you use them in the abstract, please explain. Otherwise, use day 21 and day 28 instead here.

Thanks for kind remind. We have modified it to full name according to your advice. Line 19, 23 and 25.

2.Line 46    This is a little unclear. There is very little free aroma in grape berries, most of it is bound in a precursor form. There is really no varietal aroma in grapes for that reason since we cannot perceive bound aroma compounds. This should be clarified here.

Thanks for kind remind. We have clarified it according to your advice. Line 44 and 47.

3.Line 102  This goes for the entire manuscript: please avoid abbreviations such as Fig. 1 and use Figure 1 instead.

Thanks for your suggestions. We have modified it to full name according to your advice. Line 102, 229 and 250, etc.

4.Line 105  If you randomly selected grapes for sampling, did you assess variability within the vineyard block? If not, the sample size of 1.5 Kg is too small to account for natural site variability.

Thanks for kind remind. Your question is very good. The sample at the sampling site is a variety, so it is hardly affected by the plot.

5.Line 121  This is a maceration, not an extraction in the traditional sense. 

Thanks for kind remind. We have modified it according to your advice. Line 117 and 122.

6.Line 167  How did you get to the ng/L range without pre-concentration of the analyte? 

The mass detector was operated in selected ion monitoring (SIM) mode with EI mode at 70 eV. SIM was used at mass channels of m/z 94 and 124 for IBMP. Peak areas of the ion m/z 124 were used for quantification. So it can get to the range of ng/L.

7.Line 202  Please describe the functions that you used in Excel 2019.

Thanks for kind remind. We have added it according to your advice. Line 205.

8.Line 225  How does that carbohydrate depletion work in the berry? Could it also be a mix of dehydration and microbial spoilage?

Thanks for kind remind. The samples we selected, were checked whole and plump without dehydration and microbial spoilage, and were immediately stored at 0°C after collected and immediately taken back to the laboratory. They were stored in a –80°C refrigerator before further analysis. Moreover, the precipitation in Ningxia production area is less, and there are few moldy and rotten fruits.

9.Line 235  Could this also be caused by microbial activity? Did you check for grape health at the different harvest points?

Yes, we checked for grape health at the different harvest points, so the possibility of the phenomenon caused by microbial activity less. Moreover, the precipitation in Ningxia production area is less, and there are few moldy and rotten fruits.

10.Line 273  If you are using past tense here, it sounds like you did the actual work but as far as I can see, you are referencing a paper. This needs to be a little clearer.

Thanks for kind remind. We have modified it according to your advice. Line 259, 260, and 261.

11.Line 281  Figure 1 and Figure 2 are impossible to read. Please make the heat maps bigger and the font much bigger as well.

Thanks for kind remind. We have replaced it according to your advice. Line 284 and 332.

12.Line 296  Not only the distinction of grape cultivars but also growing conditions, regions, etc.

Thanks for kind remind. As mentioned in your suggestion, growing conditions and regions are important factors affecting grapes flavor indicators. But the sample was collected at one region. Therefore, there is no such impact.

13.Line 366  Figure 3 is also hard to read. Please make the font bigger.

Thanks for kind remind. We have replaced it according to your advice. Line 369.

14.Line 390  When using abbreviations for the first time, please explain.

Thanks for kind remind. We have explained it on Line 37.

15.Line 430  This was actually already known. The aromatic qualities (and phenolic ripeness) always develops with a delay. Please elaborate on how you expect wineries to use that knowledge (or technology) in order to improve the aromatic qualities of their products.

Thanks for kind remind. We could manage the leaf curtain and posture of the cultivation process to make the sugar accumulation slower and the aroma accumulation faster, or give enough time for aroma accumulation to develop new products with higher residual sugar. Line 435 and 436. 

Reviewer 2 Report

The manuscript "GC-MS untargeting analysis of volatile compounds in four red grape varieties (Vitis vinifera L. cv) at different maturity stages near harvest" evaluates for four different grape species the indicators traditionally used to determine the ripening stage and performs an in-depth study of the aromatic components (free and bound). The results concluded that the best aromatic profile obtained for the four varieties studied was obtained after the sugar/acid indicator. Thus, the authors suggest that the determination of volatile components should be considered for harvesting in the Ningxia region.

The authors have conducted a comprehensive analysis of the basic parameters to determine the ripening stage of the grape crop and have performed an adequate statistical analysis to evaluate the differences properly.

Comments to authors:

- Line 17: Indicate what the abbreviation (3-isobutyl-2-methoxypyrazine) means.

- Line 123: Has the method indicated to perform the extraction of free volatile components been performed by the authors, or is it based on a previously published method? A bibliographic reference would be necessary, or if not, indicate in greater depth why these parameters have been selected.

- Line 202: The authors should indicate what type of statistical analysis they have carried out (ANOVA...). In addition, they should indicate what they have sought to analyze statistically (mean at different times or differences between varieties for the same parameter).

- Line 220: The authors indicate that there is a decrease in total sugar content for three of the varieties analyzed. However, when looking at Table 1, this statement cannot be confirmed since there are no statistically significant differences.

- Figure 4: Indicate at the bottom of the figure the statistics carried out in the graph.

- The authors use mostly outdated references. I would suggest modifying to more current references.

Author Response

- Line 17: Indicate what the abbreviation (3-isobutyl-2-methoxypyrazine) means.

Thanks for kind remind. We have modified it according to your advice. Line 18.

- Line 123: Has the method indicated to perform the extraction of free volatile components been performed by the authors, or is it based on a previously published method? A bibliographic reference would be necessary, or if not, indicate in greater depth why these parameters have been selected.

Thanks for kind remind. We have added it according to your advice. Line 134-135.

- Line 202: The authors should indicate what type of statistical analysis they have carried out (ANOVA...). In addition, they should indicate what they have sought to analyze statistically (mean at different times or differences between varieties for the same parameter).

Thanks for kind remind. We have modified it according to your advice. Line 219-226.

- Line 220: The authors indicate that there is a decrease in total sugar content for three of the varieties analyzed. However, when looking at Table 1, this statement cannot be confirmed since there are no statistically significant differences.

Thanks for kind remind. Indeed, there are no statistically significant differences, but there is a slight decrease in total sugar content for three of the varieties analyzed.

- Figure 4: Indicate at the bottom of the figure the statistics carried out in the graph.

Thanks for kind remind. We have added it at the bottom of the figure 4 according to your advice. Line 444-445.

- The authors use mostly outdated references. I would suggest modifying to more current references.

Thanks for kind remind. We have replaced the part of outdated references with some news.

Reviewer 3 Report

In this study four red wine grape varieties' sugars/acids ratios, free volatile compounds, bound volatile compounds, and IBMP concentration were all examined as maturation indicators. The study is well conducted but I suggest some modifications.

·         - Acronyms such as IBMP, D21 and, d28 should be explained in the Abstract.

·        - Most of the references in the Introduction section are more than 10 years old, please update them.

·        Use past sentence to describe Material and Methods.

·        Did you collect grapes of same size? How are the samples selected? I think this could be better explained

·        Line 117 high-speed crusher: For all equipment, describe model and suppler.

·        Line 120 throughout the manuscript, convert RPM in G-force.

·        Line 129 heating equipment shaker: describe the model, brand, and supplier.

·        Describe the functions that you used in Excel 2019 to perform statistical analyzes.

·        Table 1: Different letters indicate significant differences between the means (p < 0.05). Among different days? Or cultivar samples? Include information about number of samples.

·        Figure 1, 2, and 3: replace figures, they do not present the results properly, we can not read them.

·        Line 309 Include a reference

Author Response

In this study four red wine grape varieties' sugars/acids ratios, free volatile compounds, bound volatile compounds, and IBMP concentration were all examined as maturation indicators. The study is well conducted but I suggest some modifications.

 Thanks for your encouragement and advice.

  •        - Acronyms such as IBMP, D21 and, d28 should be explained in the Abstract.

Thanks for kind remind. We have explained it according to your advice. Line 18, 21, 24-25, 26.

  • - Most of the references in the Introduction section are more than 10 years old, please update them.

Thanks for kind remind. We have replaced the part of outdated references with some news.

  • Use past sentence to describe Material and Methods.

Thanks for kind remind. Yes. We have checked.

  • Did you collect grapes of same size? How are the samples selected? I think this could be better explained

Thanks for kind remind. We have added it according to your advice. Line 116.

  • Line 117 high-speed crusher: For all equipment, describe model and suppler.

Thanks for kind remind. We have added the corporate name according to your advice, however, there is no instrument model. Line 129.

  • Line 120 throughout the manuscript, convert RPM in G-force.

Thanks for kind remind. We have converted it according to your advice. Line 132.

  • Line 129 heating equipment shaker: describe the model, brand, and supplier.

Thanks for kind remind. The heating equipment shaker is an attached module of CTC.

  • Describe the functions that you used in Excel 2019 to perform statistical analyzes.

Thanks for your suggestions. It is used for data statistics and standard curve drawing.

  • Table 1: Different letters indicate significant differences between the means (p < 0.05). Among different days? Or cultivar samples? Include information about number of samples.

Thanks for your suggestions. It refers to different maturity stages. We have added it according to your advice. Line 250.

  • Figure 1, 2, and 3: replace figures, they do not present the results properly, we can not read them.

Thanks for your suggestions. We have replaced it according to your advice. Line 319, 367, 404.

  • Line 309 Include a reference

Thanks for your suggestions. We have added it according to your advice. Line 346.
